# Production and Application of Multicistronic Constructs for Various Human Disease Therapies

**DOI:** 10.3390/pharmaceutics11110580

**Published:** 2019-11-06

**Authors:** Alisa A. Shaimardanova, Daria S. Chulpanova, Kristina V. Kitaeva, Ilmira I. Abdrakhmanova, Vladislav M. Chernov, Catrin S. Rutland, Albert A. Rizvanov, Valeriya V. Solovyeva

**Affiliations:** 1Institute of Fundamental Medicine and Biology, Kazan Federal University, 420008 Kazan, Russia; aliceshaimardanova@mail.ru (A.A.S.); daryachulpanova@gmail.com (D.S.C.); 2019olleth@mail.ru (K.V.K.); ilma25.94@mail.ru (I.I.A.); chernov@kibb.knc.ru (V.M.C.); rizvanov@gmail.com (A.A.R.); 2Shemyakin-Ovchinnikov Institute of Bioorganic Chemistry, The Russian Academy of Sciences, 117997 Moscow, Russia; 3Kazan Institute of Biochemistry and Biophysics, FRC Kazan Scientific Center of RAS, 420111 Kazan, Russia; 4School of Veterinary Medicine and Science, University of Nottingham, LE12 5RD Nottingham, UK; catrin.rutland@nottingham.ac.uk

**Keywords:** 2A peptides, IRES, multicistronic constructs, gene therapy, neurodegenerative diseases, metabolic diseases, cancer, autoimmune diseases, cardiovascular diseases

## Abstract

The development of multicistronic vectors has opened up new opportunities to address the fundamental issues of molecular and cellular biology related to the need for the simultaneous delivery and joint expression of several genes. To date, the examples of the successful use of multicistronic vectors have been described for the development of new methods of treatment of various human diseases, including cardiovascular, oncological, metabolic, autoimmune, and neurodegenerative disorders. The safety and effectiveness of the joint delivery of therapeutic genes in multicistronic vectors based on the internal ribosome entry site (IRES) and self-cleaving 2A peptides have been shown in both in vitro and in vivo experiments as well as in clinical trials. Co-expression of several genes in one vector has also been used to create animal models of various inherited diseases which are caused by mutations in several genes. Multicistronic vectors provide expression of all mutant genes, which allows the most complete mimicking disease pathogenesis. This review comprehensively discusses multicistronic vectors based on IRES nucleotide sequence and self-cleaving 2A peptides, including its features and possible application for the treatment and modeling of various human diseases.

## 1. Introduction

Recently, gene therapy was introduced as having significant potential for the treatment of various diseases. Gene therapy involves the modulation of target gene expression to achieve a therapeutic effect [1]. From the beginning, gene therapy was mainly based on viral vectors, which met a lot of skepticism in both the scientific and medical communities because of the unexpectedly high toxicity of viruses [2]. However, with the advent of various gene therapeutic agents for the treatment of cancer [3], blindness [4], immune [5] and neuronal disorders [6], renewed attention has been focused on gene therapy. Many of the currently existing replication-defective viruses can carry large therapeutic genes, efficiently transduce various types of cells, and provide long-term and stable expression of genes of interest [7]. However, viral vectors can also be immunogenic or pathogenic, or can cause inflammation or malignant transformation of infected cells [8].

Other non-viral approaches for gene delivery, such as cationic lipids (liposomes and lipoplexes) and polymers (polyplexes), have their advantages and disadvantages [9,10]. Non-viral delivery methods are safer, lower immunogenic responses, as well as allowing the downloading and delivery of bioactive molecules and genetic material of unlimited size [11].

Besides the question of delivery, gene therapy also faces challenges that require the delivery and expression of several therapeutic agents at once. Cloning of several genes into one vector greatly facilitates the implementation of many fundamental tasks, and the joint expression (co-expression) of many genes improves the effectiveness of gene therapy. Examples of this include co-expression of a gene of interest with a reporter gene for the rapid identification and selection of modified cells [12], co-expression of several subunits of one protein [13,14], several proteins, for example, for the delivery of enzyme complexes [15,16], and transcription factors [17,18], or several therapeutic agents in combined gene [19,20] or gene and cell therapy [21,22,23,24,25]. Co-expression of several genes in one vector is also used to create animal models of various hereditary diseases caused by mutations in several genes rather than just one. These vectors provide expression of all mutant genes, which allows mimicking disease pathogenesis to the greatest possible extent [26,27]. Various molecular strategies are used for the simultaneous expression of several genes in one vector, including the use of multiple promoters [28], signals of splicing [29], fusion of genes [30] and cleavage factors [31,32]. The use of multiple promoters suggests an existence of several independent transcription units, each of which has its own promoter and open reading frame (ORF) [33]. The use of different promoters is suitable when simultaneous but not sequential gene expression is important. These kinds of plasmids containing several genes under different promoters are also used in gene expression profiling studies [34]. However, the combined use of multiple promoters can result in the transcriptional interference of the downstream promoter and the suppression or stimulation of the transcriptional process [35].

Another strategy is to insert splicing signals before and after the first gene of interest. As a result, several mature mRNAs can be produced by splicing a single pre-mRNA. There are several transcriptional units in this variant, which provide independent gene expression; however, there are risks of splicing and rearrangement changes during vector construction [36]. Such vectors have their advantages; for example, there is an alternative-splicing-based bicistronic vector in which the ratio between the synthesized proteins can be controlled by changing the consensus sequence. This vector has been successfully used for the production of recombinant monoclonal antibodies [29].

Gene fusion is mostly used to attach the gene of interest with a reporter gene [37]. In this case, the expression of genes is controlled by one promoter and results in a hybrid protein, which may affect its functions [38]. Insertion of proteolytic cleavage sites between genes is also used. The use of this strategy is limited however, because the protease and protein have to be localized in the same subcellular space [31].

Among all of the strategies used to construct vectors containing two or more genes, the most promising and widely used approaches presently are the use of internal ribosome entry site (IRES) and self-cleaving 2A peptides, since these methods can overcome many of the disadvantages occurring in the above strategies. Vectors encoding the nucleotide sequences of IRES or 2A peptides are called multicistronic vectors, since they simultaneously express two or more separate proteins from the same mRNA unlike other vectors [12]. This review discusses the mechanisms of IRES-based and 2A peptide-based protein synthesis and their features and compares them with each other. The perspectives about the use of IRES and 2A peptides to create multicistronic vectors for gene and gene and cell therapy and the prevention of various human diseases are also discussed.

## 2. IRES

It is known that protein translation can occur cap-dependently or cap-independently. Most often, in the initiation of translation in eukaryotes, the cap which binds to the small (40S) subunit of ribosome and many translation initiation factors (eIF) are required [39,40]. The IRES sequence controls cap-independent protein synthesis. The IRES-dependent initiation of the translation occurs independently of the mRNA 5’-end and is therefore called an internal initiation of translation (Figure 1). The IRES sequence, like the 2A peptides, was first discovered in viruses [41]. IRES was initially found to translate viral proteins; however, it was later determined that the initiation of translation of some cellular genes can also be IRES-dependent [42,43,44].

The IRES-dependent initiation of translation in various viruses occurs in different ways. Due to their specific structures, some IRESs can functionally replace many protein factors and, it is also assumed, can bind and modify the ribosome conformation [45,46], which is why the IRES-dependent initiation of translation does not require a complete (or any) eIF set. The eIF set depends on the structure of IRES that varies in differing types of viruses [46]. All of the IRES types have complex secondary and tertiary structures. IRES is a hinge molecule whose constituent elements can move relative to each other. Conformational changes in IRES are necessary for the formation of the initiation complex; the greater the need for additional initiation factors, the more flexible the molecule [47]. The secondary structure of IRES from different viruses is comprehensively discussed in a previous publication [45].

IRES is widely used to transfer several genes in multicistronic vectors, since it allows the expression of several proteins from a single transcription unit. In the multicistronic vector, the first gene upstream of IRES is translated in the cap-dependent manner, and in the downstream gene, translation initiation is IRES-dependent (Figure 1). One of the significant disadvantages of such constructs is a decreased IRES-dependent expression of the second gene [48]. The expression level of all IRES-dependent proteins in one multicistronic vector is the same, regardless of the order of the genes. The translation of all genes which are located downstream of the IRES occurs independently of each other [49], in contrast to vectors containing 2A peptide sequences [32]. A significant advantage of IRES sequences is the complete separation of proteins, whereas the use of 2A peptides can result in incomplete cleavage of the translated polypeptide (Table 1) [50]. 

## 3. Self-Cleaving 2A Peptides

Self-cleaving 2A peptides are promising candidates for the production of multicistronic vectors due to their small size and self-cleavage ability. 2A peptides are composed of 16–20 amino acids and originate from viral RNA. The most common 2A peptides used to produce multicistronic vectors are F2A (2A peptide derived from the foot-and-mouth disease virus), E2A (2A peptide derived from the equine rhinitis virus), P2A (2A peptide derived from the porcine teschovirus-1), and T2A (2A peptide derived from the Thosea asigna virus) [51].

The mechanism of 2A peptide cleavage is also of interest. 2A peptides were first discovered in 1991 in the foot-and-mouth disease virus belonging to the Picornaviridae family. At that point, it was suggested that the peptide could be cleaved by a protease encoded by the virus or an infected cell; however, it later became clear that proteases are not involved in this process, and the cleavage is mediated via a ribosomal skip mechanism [52,53]. It is known that proteins of picornaviruses are encoded in one long open reading frame (ORF). The cleavage sites of these peptides are located between the C-terminal glycine of 2A and the N-terminal proline of 2B (aminoacidic sequences of 2A peptides are represented in a previous publication [54]). This process takes place inside the ribosome during protein synthesis, the formation of a normal peptide bond between these amino acids at the cleavage site is inhibited, but the cleavage does not affect the translation of the subsequent protein, since ribosome dissociation does not occur (Figure 1) [55]. However, this mechanism can fail, in that ribosomal skipping sometimes does not occur and the protein synthesis continues which leads to the formation of fused (uncleaved to each other) proteins. Sometimes, when the synthesis of a gene located upstream of the 2A peptide sequence is completed, the ribosome drops off and translation stops, and eventually only the first protein is synthesized. Despite such failures, 2A is the best way to provide co-expression of several genes compared with other strategies [32].

Because of the incomplete cleavage of the products, the 2A-based multicistronic cassette can lead to a decreased yield of the final product and to a protein dysfunction, and it therefore becomes necessary to develop various ways in which to improve the cleavage of such structures. The formation of fused products is believed to be due to the inhibition of the 2A cleavage reaction. The use of elongated 2A with the additional spacers in the N-terminus of 2A peptide sequence can provide a solution, since it leads to a decrease in the inhibition rate of the 2A cleavage reaction [56]. The insertion of additional spacer sequences as well as the furin cleavage site are used for this purpose [57]. A glycine–serine–glycine (GSG) spacer flexible linker sequence is most commonly used for the best cleavage [58,59,60]. In addition, it was found that the order of various 2A peptide sequences in multicistronic constructs can also affect the cleavage efficiency [32].

Liu et al. investigated the levels of gene expression downstream and upstream of the 2A peptide sequence and the efficiency of protein cleavage in multicistronic vectors with different combinations of 2A peptides [32]. The authors showed a decrease in protein expression levels in the second position in 2A peptide-based bicistronic constructs. Interestingly, in tricistronic constructs the highest expression is observed in the first position, then in the third, and the lowest expression in the second position, although a gradual decrease in the amount of translated protein along the multicistronic vector is expected. Researchers suggest that this may be due to the fact that T2A was used before the third gene, and the P2A peptide sequence was used before the second gene. Thus, the use of different 2A peptides can influence the expression levels of downstream proteins. A gradual decrease in the gene expression from the first to the last is also observed in quadricistronic constructs. The authors observed that the combination of 2A peptide sequences in the following order, namely T2A, P2A, and E2A, was optimal when creating multicistronic vectors containing four genes [32]. Kim et al. compared the cleavage efficacy of P2A, T2A, E2A, and F2A in different cell cultures, such as human cell lines, zebrafish embryos, and adult mice liver. In all cell cultures, P2A showed the best cleavage efficiency [54].

The efficacy of using different 2A peptides for the production of monoclonal antibodies in the Chinese hamster ovary (CHO) cell line was investigated by Yang et al. [57]. For this purpose, constructs encoding the genes of the heavy and light chains of monoclonal antibodies, separated by different 2A peptide sequences (F2A, E2A, P2A, and T2A) were produced. The use of such structures is important for controlling the ratio of heavy and light chains. Different 2A peptides have shown differing cleavage efficiencies and, consequently, differential monoclonal antibody yields, since the uncleaved polypeptide products were removed. It has been shown that the use of any 2A peptide does not lead to complete cleavage, but T2A showed the highest cleavage efficiency and level of product expression. The GSG linker has also been shown to significantly increase 2A cleavage efficiency [58]. A study of the efficacy of 2A cleavage in Drosophila cell culture has also confirmed that T2A and P2A cleave most efficiently [61].

Thus, studies aimed at comparing the cleavage efficiency of 2A peptides have shown different results, which can be associated with many factors, including experimental conditions and the use of various cell cultures. However, most often the best result is shown by T2A and P2A [58,59,60,61]. In order to identify the most suitable type of 2A for certain tasks and select the optimal conditions for using each 2A, a comprehensive comparison and more detailed investigations of 2A are required.

The comparison of gene expression dynamics in multicistronic vectors containing IRES and P2A peptide sequence has shown that P2A-based constructs provide more correlated gene expression than IRES-based constructs in HEK293 cells (Table 1); however, no significant correlation was observed in Neuro2a cells. These results confirm that IRES activity may vary depending on the cell type [62].

## 4. Multicistronic Vectors for Neurodegenerative Disease Therapy

Neurodegeneration is characterized by the disturbance of structure and the loss of neuron function. Many neurodegenerative disorders are caused by the accumulation and distribution of abnormal protein aggregates in the central nervous system (CNS) [63]. Gene therapy may help overcome the challenges of neurodegenerative disease treatment, including difficulties with the delivery and distribution of therapeutic agents in the CNS, and multicistronic vectors enable the simultaneous delivery and expression of several therapeutic genes.

The greatest success in the use of multicistronic vectors has been achieved in the treatment of Parkinson’s disease (PD) characterized by selective loss of dopaminergic neurons in the *substantia nigra pars compacta* (SNc), which leads to complete dopamine depletion and serious impairment of motor functions (tremor, rigidity, akinesia, and other similar symptoms). Since the dopamine replacement strategy is the most perspective strategy to treat PD [64], a lentiviral multicistronic construct encoding genes of three proteins essential for dopamine biosynthesis, namely tyrosine hydroxylase (TH), aromatic amino acid dopa decarboxylase (AADC) and GTP cyclohydrolase 1 (CH1) linked through IRES has been proposed for restoring dopamine synthesis. Injection of this construct into the striatum of a PD model (the 6-OHDA-lesioned rat striatum) in mice has led to the efficient transduction and long-term gene expression, catecholamine production, and a significant decrease of apomorphine-induced motor asymmetry [65]. In the same way, a bicistronic retroviral construct containing human TH and rat GC genes, separated by IRES, and necessary for the synthesis of l-DOPA, the dopamine biosynthesis intermediate product (3,4-dihydroxyphenylalanine), was produced. The efficiency of synthesis was confirmed in mesenchymal stem cell (MSC) culture in vitro [66].

An experimental drug, ProSavin, can be considered the most efficient for the treatment of PD and is intended for administration into the striatum of the patient’s brain. The drug is a viral vector encoding AADC, TH, and CH1 genes, separated by IRES. Phase I/II clinical trials (numbers NTC00627588 and NCT01856439; EudraCT numbers: 2007-001109-26 and 2009-017253-35) have shown that ProSavin has a favorable safety profile and results in the improvement in the defined “off” Unified Parkinson’s Disease Rating Scale part III motor scores after the injection into the striatum of PD patients [67].

Another therapeutic approach is the delivery of trophic factor genes, such as brain-derived neurotrophic factor (BDNF), which has a neuroprotective function and regulates the transmission in dopaminergic neurons in SNc, protecting SNc from 6-hydroxydopamine (6-OHDA). Delivery of the BDNF gene as part of an adeno-associated virus (AAV)-based bicistronic vector co-expressing the reporter gene of green fluorescent protein (GFP) through IRES resulted in a stable co-expression of BDNF and GFP in SNc for nine months in vivo. Higher indices of motor functions (locomotor activity and rotational activity) were observed in animals treated with BDNF compared to controls; however, BDNF failed to affect nigrostriatal dopaminergic survival [68].

Some successes have also been achieved in the development of multicistronic vectors for the treatment of amyotrophic lateral sclerosis (ALS) characterized by cholinergic dysfunction. The injection of human umbilical cord blood mononuclear cells (hUCB-MCs) modified with an adenovirus encoding vascular endothelial growth factor (VEGF) and basic fibroblast growth factor (FGF2) linked through P2A peptide sequence into an ALS mouse model led to the secretion of both growth factors in the spinal cord, as well as the absence of immune response to 2A antigen after 28–52 days. Meanwhile, the viability of xeno-transplanted cells remained at the same level in the spinal cord for one month [51].

Multicistronic vectors can also be used to produce transgenic animals for modeling human neurodegenerative diseases, for example, Alzheimer’s disease (AD). Application of a retroviral vector containing the amyloid precursor protein (APP), Tau protein (TAU), and human presenilin 1 (PS1) genes with six well-characterized mutations, separated by 2A peptide sequence has allowed the production of transgenic pigs with a high level of expression of APP, TAU, and PS1 in the animal brain [26]. However, further studies are required to confirm the development of the pathogenesis of AD in transgenic animals at the cellular, tissue, and body levels. It is of interest to create multicistronic vector-based test systems for screening potential compounds for the treatment of neurodegenerative diseases. An example of such a test system is the human neuroblastoma cell line co-expressing APP and TAU genes separated by IRES and designed for the screening of AD therapy drugs [69,70]. Using the above test system, curcumin, its structural analogue demethoxycurcumin [69], and memantine (non-competitive NMDA receptor antagonist (*N*-methyl-d-aspartate)) [70] have been able to inhibit APP and TAU IRES-dependent translation initiation.

## 5. Multicistronic Vectors for Metabolic Disease Therapy

Metabolic diseases are characterized by the alteration of metabolic processes most often due to the lack or deficiency of enzyme activity. The use of multicistronic vectors for the production of insulin-producing cells has shown promising results in the development of new methods for diabetes treatment [71]. Three transcription factors are required for insulin production—pancreatic and duodenal homeobox-1 (Pdx1), neurogenin 3 (Ngn3), and v-musculoaponeurotic fibrosarcoma oncogene homolog A (MafA). The transfection of hepatocytes with a multicistronic vector carrying the Pdx1, Ngn3, and MafA genes has allowed the reprogramming of hepatocytes into insulin-producing cells in vitro and correcting the diabetic state in vivo [32]. The use of insulin-producing induced pluripotent stem cells (iPSCs) for the treatment of type I diabetes is also under investigation. To differentiate iPSCs into insulin-producing cells, iPSCs were co-transduced with multicistronic adenoviral vectors containing the Pdx1, Ngn3, or MafA genes linked through IRES to the green fluorescent protein (GFP) reporter gene. The resulting insulin-producing cells were transplanted into the liver parenchyma of type I diabetic mice, which resulted in hyperglycemia reversal [72]. The clinical manifestation of the disease in some metabolic syndromes occurs only when enzyme activity significantly decreases; often 5%–25% of the normal activity of the enzyme is sufficient to maintain the patient’s healthy state [73,74]. One of these diseases is GM2-gangliosidosis (Tay-Sachs disease, Sandhoff disease, and AB variant), which is a part of the group of lysosomal storage diseases caused by the accumulation of GM2 gangliosides due to the deficiency of the β-hexosaminidase (Hex) enzyme. The accumulation of GM2 gangliosides primarily affects CNS functions, resulting in severe neurodegeneration [74]. The use of AAV9 encoding the genes of the α- and β-subunits of the HexA enzyme (HEXA and HEXB genes, respectively) linked through P2A peptide sequence has been reported. The systemic administration of this construct into a murine model with Sandhoff disease has increased animal survival rates by 56%, due to the fact that the bicistronic construct provides maximum overexpression and secretion of the HexA enzyme and AAV9 is able to overcome the blood–brain barrier [13]. 

The use of multicistronic vectors for the treatment of Fabry disease, an X-linked inherited disorder characterized by the deficiency of the α-galactosidase A (a-Gal A) enzyme that leads to the impaired metabolism of sphingolipids, has also been evaluated. The use of bicistronic retroviral vectors expressing a-Gal A gene and drug-selectable multidrug resistance gene 1 (MDR1), which are separated by the IRES sequence, has shown a significant increase in the activity of the missing a-Gal A enzyme in cell culture in vitro [75].

The greatest success has been achieved in the treatment of mucopolysaccharidosis IIIA, a progressive hereditary neurodegenerative disorder that manifests in early childhood. The disease is caused by the mutations in the gene encoding heparan-*N*-sulfamidase and *N*-sulfoglycosamine sulfohydrolase (SGSH) lysosomal enzymes, resulting in the accumulation of heparan sulfate. Sulfatase-modifying factor (SUMF1) is an activator of the SGSH catalytic site. Studies in murine models have shown that intraparenchymal administration of AAVrh.10-SGSH-IRES-SUMF1 improves heparan sulfate catabolism and decreases microglia activation at injection sites, but no improvement has been observed in areas of the brain distant from the injection site. Based on the obtained results, phase I/II clinical trials were conducted, during which four children were intracerebrally injected with AAVrh.10-SGSH-IRES-SUMF1. As a result, a cerebral atrophy was stopped in two of the four children, and an improvement in behavior, attention, and sleep was observed in three patients [76].

## 6. Multicistronic Vectors for the Treatment of Autoimmune Diseases

Autoimmune diseases are multifactorial disorders associated with the impairment of immune system functions, including the abnormal regulation of cytokines, which are good targets for gene therapy [77]. It is known that T-helper 17 (Th17) cells producing interleukin (IL)-17 play a key role in inflammation initiation in autoimmune diseases, whereas IL-27 controls autoimmune diseases by suppressing Th17 from producing IL-17. IL-27 also promotes the differentiation of T-cells that secrete IL-10, which has anti-inflammatory properties. It has been suggested that IL-27 may serve as an additional agent in the therapy of many autoimmune diseases. MSCs transduced with a lentivirus encoding two IL-27 subunits (p28 and EBI3) linked through IRES have been used for cell-mediated gene therapy of autoimmune diseases. It is worth noting that IRES-dependent expression of the EBI3 subunit was decreased by one third in comparison to p28 subunit expression, but this did not affect the functionality of the resulting protein. However, the therapeutic effects of the construct have not been evaluated [78].

Anti-inflammatory cytokine-based therapy may be a promising approach for the treatment of multiple sclerosis, a chronic autoimmune disease in which demyelination of neurons in the CNS occurs, which leads to neurological disabilities [79]. IL-4, leukemia inhibitory factor (LIF), and IL-10 are anti-inflammatory cytokines, whereas IL-17 is a pro-inflammatory cytokine [80,81,82,83]. It has been suggested that inhibition of pro-inflammatory cytokines and overexpression of anti-inflammatory cytokines may be an effective approach in the gene therapy of autoimmune diseases. A lentivirus based on a multicistronic vector encoding IL-4, IL-10, and LIF through T2A and P2A peptide sequences was used to modify Wharton’s jelly stem cells, the efficacy of which was further investigated in the experimental model of mice with autoimmune encephalomyelitis. As a result, the damage in the brains of a mouse model, the amount of IL-17, and cell infiltration in the brain were decreased in vivo [79].

## 7. Multicistronic Vectors for Cardiovascular Disease Therapy

Some studies have shown the efficacy of multicistronic vectors in the treatment of cardiovascular diseases. For example, an IRES-based plasmid vector that encodes two angiogenic molecules, FGF2 and the cysteine-rich angiogenic inducer 61 (Cyr61), has been found to be more effective in stimulating the formation of a new vascular network in the hindlimb ischemia mouse model compared to monocistronic construction, despite the fact that bicistronic vectors express a smaller amount of the corresponding protein molecules than vectors encoding one gene [84]. This data indicate the possibility of increasing therapy effectiveness, while reducing the side effects caused by the expression of a large amount of one protein. A bicistronic IRES-based AAV vector encoding VEGF and bone morphogenetic protein (BMP) genes was used for the genetic modification of rabbit bone marrow-derived mesenchymal stem cells. The synergistic effect of the expression of two proteins with angiogenic and osteogenic functions allowed the simultaneous stimulation of bone formation and vascular regeneration in the rabbit hindlimb ischemia model in vivo [85]. Other IRES-based multicistronic vectors (VEGF-A/FGF4 and VEGF165/stromal cell-derived factor-1 (SDF-1)) have also been shown to be effective in therapy for ischemia animal models in vivo [86,87].

Simultaneous equivalent expression of genes with a 2A peptide sequence-based vector encoding transcription factors Gata4, Mef2c, and Tbx5, which are necessary for the in situ reprogramming of cardiac fibroblasts in induced cardiomyocytes, has shown a higher efficacy in improving ventricular function in mice after in vivo coronary ligation compared to monocistronic construction [18]. Clinical trials of bicistronic plasmid encoding VEGF and FGF genes through IRES have shown plasmid safety, improvement in tolerance to both physical activity and clinical symptoms, but not improvement in myocardial perfusion [88]. However, the potential of multicistronic vectors for the treatment of cardiovascular diseases remains largely unexplored. 

## 8. Multicistronic Vectors for Cancer Therapy

Since the etiology of cancer is fairly diverse, the treatment of cancer requires the development of complex approaches, including immunotherapy, where the combination of various therapeutic agents can significantly increase the effectiveness. One of the first examples of the use of multicistronic constructs in cancer treatment was the production of an adenoviral vector encoding two subunits of human IL-12 through IRES. The expressed cytokine has shown all biological activities of recombinant human IL-12, such as the stimulation of proliferation of the activated natural killer (NK) and T-cells, induction of the interferon-gamma (IFN-γ) secretion by NK and T-cells, and enhancement of lymphokine-activated killer (LAK) activity [89]. There are other constructs simultaneously encoding several cytokines [90,91]. For example, co-expression of IL-2 and tumor necrosis factor (TNF-α), linked by IRES, in an oncolytic adenovirus genome, has allowed increases in the number of CD4^+^ and CD8^+^ tumor-infiltrating lymphocytes, as well as T-helper 1 (Th1) and CD86^+^ dendritic cells (DCs) in vivo [90,92]. Besides the immunomodulatory effect, higher direct antitumor activity can be achieved by using IRES. For example, the combination of endostatin (ES) and herpes simplex virus thymidine kinase (HSV-TK) suicidal genes has significantly inhibited cell proliferation and induced apoptosis in C6 glioma cells compared to individual agents [93]. IRES-based multicistronic vectors containing the combination of suicidal gene and immunomodulating cytokine have also been described. Such combinations do not lead to a synergistic enhancement of the antitumor effect in vitro; however, the direct immunomodulatory effect has not been evaluated [94]. Multicistronic vectors have also been used to inhibit tumor lymphangiogenesis and angiogenesis. Thus, an adeno-associated vector with fibstatin and platelet factor-4 variant 1 (CXCL4L1) genes has inhibited angiogenesis, lymphangiogenesis and tumor spread to the lymph nodes in vivo [95].

The number of studies on the development of multicistronic structures with 2A peptide sequences for cancer treatment is lower than with IRES, but judging by the number of publications in recent years, this approach is also being actively studied. One of the first studies with 2A-peptide based vectors was the production of the zaptuximab chimeric antibody, which binds to a receptor responsible for selective induction of apoptosis in tumor cells. The lentiviral vector encoding zaptuximab heavy and light chains through the F2A peptide sequence has been shown to induce apoptosis in various tumor cell lines in vitro and in the mouse colon cancer model in vivo [96]. The F2A peptide was also used to express anti-carcinoembryonic antigen (CEA)×anti-CD3 di-antibodies, and the assembly efficiency of the antibodies was 3 times higher than IRES when using F2A, which led to an increase in T-cell cytotoxic response against CEA-positive tumor cells [97]. The expression of yeast cytosine deaminase (yCD2) from a retroviral genome through T2A has also been increased compared to IRES-based expression. The use of the construction with the T2A peptide led to a higher survival rate of model mice with intracranial tumors [59]. The assembly efficiency of IL-12 has also been increased by at least 2 times due to the P2A peptide sequence compared with IRES, which led to a significant increase in CD8^+^ T-cell number into the tumor environment and an increase in systemic anti-tumor response [98]. The use of 2A peptides in cancer treatment so far is reduced due to an increase in the production of chimeric antibodies or proteins consisting of several subunits. The prospect of combining various therapeutic agents in one vector remains to be investigated.

## 9. Multicistronic Vectors for the Prevention of Viral and Bacterial Infections

The use of multicistronic vectors in the field of infectious diseases is mainly focused on the development of preventive vaccines. Simultaneous co-expression of several viral antigens can provide more effective immunization. For example, immunization of mice with a bicistronic vector expressing two highly conserved antigens of influenza virus, nucleoprotein (NP) and matrix (M1), separated by IRES has resulted in a full protection against the disease [99]. The same IRES-based strategy has been used to create a vaccine against *Staphylococcus aureus*-induced mastitis [100], the immunization with which has reduced the incidence of mastitis in cows. IRES has also been used to create antibodies against staphylococcal enterotoxins (SEs), but its therapeutic effect has not been evaluated to date [101]. IRES is also used to create vectors encoding antigens and an immunomodulating agent to modulate immunogenicity. The use of a bicistronic vector encoding Hc gene of botulinum neurotoxin serotype A (BoNT/A) and interleukin 4 (IL-4) gene via IRES has significantly increased the anti-AHc antibody titers, serum neutralization titers, and survival rates of immunized mice [102].

The 2A peptide is also used to produce antiviral vaccines; for example, recombinant modified vaccinia Ankara vector, encoding influenza A NP, M1, and PB1 protein genes, through P2A peptide sequence has shown a complete protective effect when challenged with 2 LD50 of A/California/7/09 virus [103]. In general, the use of the strategy for creating multicistronic structures has not yet found a wide distribution in the prevention and the treatment of infectious diseases. The number and published data from existing investigations offer little hope for rapid development in this area in the near future.

## 10. Conclusions and Future Perspectives

IRES-based and 2A peptide-based vectors are promising tools that can be used in combined gene therapy of many rare and/or heterogeneous diseases. The current number of studies investigating multicistronic vectors is relatively small, but the data now available have shown the opportunities available to combine therapeutic agents in order to increase effectiveness due to synergistic effects. In addition, the clinical trials published to date have indicated the safety of the use of multicistronic vectors in human therapy, which offers great opportunities for the development of new constructs. However, to successfully use multicistronic vectors in the treatment of various diseases, it is necessary to consider the major shortcomings of both IRES-based and 2A peptide sequence-based vectors, namely, the relatively low expression of the second protein in the IRES-based constructs and the problems with the polypeptide cleavage in 2A peptide-based constructs. Incomplete cleavage of products of 2A-based multicistronic constructs can lead to a decreased yield of the final product and to the disruption of the protein functions. Moreover, existing approaches to increase the efficiency of cleavage significantly raise the efficiency, but still do not enable complete cleavage. In addition, the unequal expression of proteins located in the different parts of the vector or separated with different 2A peptide sequences forces us to pay more attention to the order of genes in multicistronic vectors in order to achieve more significant therapeutic effects.

The use of multicistronic vectors has found the greatest success in the treatment of neurodegenerative diseases. In particular, the experimental drug ProSavin, encoding AADC, TH, and CH1 genes, separated by IRES, has shown safety and promising efficacy in the therapy of PD patients in phase I/II clinical trials. Moreover, 2A-based multicistronic vectors can be used to produce transgenic animals for modeling human AD. Encouraging results have also been obtained in the treatment of metabolic diseases, particularly in the treatment of GM2-gangliosidoses and the reprogramming of hepatocytes or iPSCs into insulin-producing cells for diabetes therapy. Moreover, phase I/II clinical trials of IRES-based multicistronic constructs have shown the efficacy in the treatment of mucopolysaccharidosis IIIA. The use of IRES- and 2A-based multicistronic vectors for the treatment of cardiovascular diseases is limited to a small number of in vitro and in vivo studies. Therefore, it is difficult to judge the development prospects of using multicistronic constructs in this field. This is not the case with cancer therapy, where there are a large number of publications in recent years. Although the use of IRES- and 2A-based multicistronic vectors in cancer therapy is currently limited to in vitro and in vivo studies, and the use of 2A peptides is limited to the production of chimeric antibodies or proteins consisting of several subunits, we believe that the number of investigations in this area will continue to grow. In the field of infectious diseases, multicistronic constructs have so far found use only in the production of preventive vaccines and progress in this area is not expected in the near future.

Multicistronic vector-based therapy has not yet revealed its potential and requires further research, as well as the development of new approaches to increase the efficiency of cleavage in 2A-based constructs. New developments in this area will significantly increase efficiency and expand the scope of multicistronic structures in the therapy of various diseases.

## Figures and Tables

**Figure 1 pharmaceutics-11-00580-f001:**
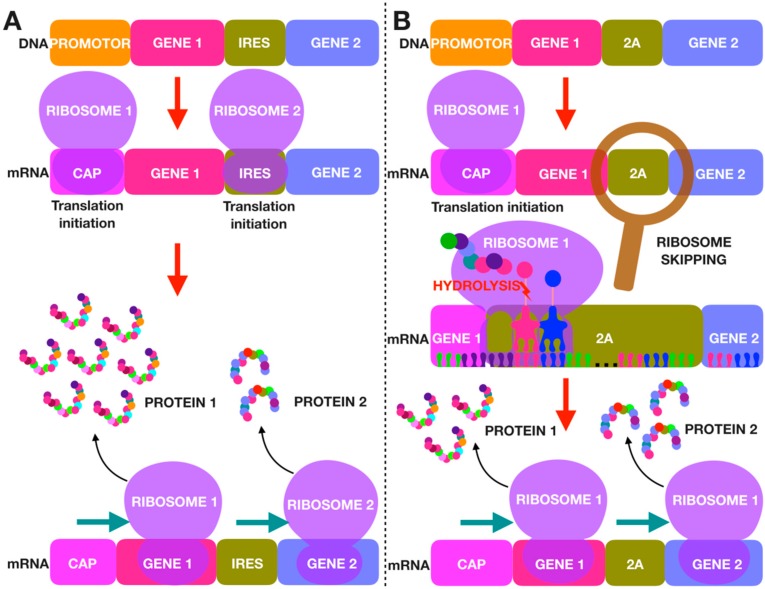
Comparison of the translation mechanism in internal ribosome entry site (IRES)-based and 2A peptide-based multicistronic constructs. (**a**) In IRES-based constructs, translation initiation of each gene occurs independently. IRES sequence controls cap-independent protein synthesis. The IRES-dependent initiation of the translation occurs independently of the mRNA 5’-end. In the IRES region (like in the cap region), ribosome binds with mRNA. The translation of the gene located upstream of the IRES is cap-dependent, and that of the gene located downstream of the IRES is IRES-dependent. Thus, the translation of two proteins located at the same mRNA is completely separated. (**b**) In constructs with 2A peptide sequences, translation is initiated once, synthesis along mRNA occurs continuously; however, during translation, the first peptide breaks from the second peptide in the 2A region. 2A peptide cleavage site is located between glycine and proline. The cleavage process occurs inside the ribosome during protein synthesis, the formation of a normal peptide bond between the amino acids is inhibited only at the cleavage site, thus the cleavage does not affect the translation of the subsequent protein, synthesis continues without the dissociation of ribosome. Theoretically, ribosomal skipping provides complete cleavage of the final product; however, mechanism failures can occur. Ribosomal skipping may not occur due to the inhibition of the cleavage reaction; in this case, fused products are formed, or after the synthesis of the first polypeptide, the ribosome can drop off and then only the first protein is synthesized.

**Table 1 pharmaceutics-11-00580-t001:** Comparison of the characteristics of various strategies for the expression of multiple genes.

Characteristic	Properties of Protein Synthesis	Size	Gene Expression Level	Cleavage
**IRES**	Ribosome dissociates when the synthesis of the first gene is complete, the synthesis is interrupted, a new translation initiation complex is assembled in the IRES region	Large (over 500 bp)	The translation efficiency of the gene located downstream of the IRES is much lower than that of the gene located upstream of the IRES	The resulting proteins are always separated from each other
**2A peptides**	Continuous synthesis of two proteins. Ribosomal skipping occurs after the synthesis of the first gene is complete, synthesis continues without the dissociation of ribosome	Small (54–66 bp)	Better correlation of the expression of genes placed upstream and downstream of the peptide sequence	Incomplete digestion of protein products is possible
**Multiple promoters**	Two separate transcription units, completely separated synthesis	Large	Poor correlation of the expression of two genes	Independent products
**Splicing signals**	Two separate transcription units, completely separated synthesis	Small	Uncertain correlation of the expression of two genes	Independent products
**Fusion of genes**	One chimeric polypeptide is translated, which can lead to impaired function	No intermediate sequences	Guaranteed co-expression	Not cleaved product
**Cleavage factors**	Two proteins are translated together	Small	Guaranteed co-expression	Cleaved by cellular proteases after protein synthesis

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
