# Peer review of "Production and Application of Multicistronic Constructs for Various Human Disease Therapies"

_pharmaceutics, 2019, doi:10.3390/pharmaceutics11110580_

Round 1

Reviewer 1 Report

This topic of this review might be interesting for the reader in this field. Authors introduced two popular strategies for multigene expression, and discussed those two application in some specific disease contexts. The information overviewed here covers a broad spectrum. However, some important contexts are still missing in this review, which prevent this review from being an inspirational recap to this topic. Authors have to address the below concern to further improve the quality of this review.

1 The introduction section should be strengthened. The beauty, importance and urgency as well as present status of gene therapy should be included.

2 The other strategies for coexpression, such as multipromoter, splicing signaling, fusion, etc., should also be included and compared along with 2A peptides and IRES. A table head-to-head comparison is highly recommended.

3 Some necessary schematic depicting 2A peptides and IRES should be provided.

4 In the section of “3. Self-cleaving 2A peptides”. Different 2A peptides sequence used in vector constructor should be summarized and compared.

5 Some novel retroviral replicating vectors involved in gene delivery should be also overviewed here. As a hot topic, inclusion of some non-viral gene delivery vector, like nanoparticles-based strategy, will greatly broaden the spectrum of this review, which might cover more reader’ interests.

6 Under section 4 to 8, to attract reader’s attention, some figures with interesting results and important finding in those literatures can be listed, instead of only using plain text.

7 Multicistronic vectors for other diseases field should also be mentioned. Such as infectious diseases,

8 Line 326: [81]. “The use of 2A peptides in cancer treatment so far is reduced to an increase in the production of”. There is a grammar error in this sentence. “so far is reduced due to” not “so far is reduced to”. Authors should verify on that.
9 Abbreviation list should be included.

Author Response

We would like to thank Reviewer for their valuable comments which have helped us to significantly improve our manuscript. The followings are our point-by-point responses (your comments are in bold text and our responses are in ordinary type):

1. The introduction section should be strengthened. The beauty, importance and urgency as well as present status of gene therapy should be included.

We have added the information about gene therapy, history of its development and current status (Lines 33-42).

2. The other strategies for coexpression, such as multipromoter, splicing signaling, fusion, etc., should also be included and compared along with 2A peptides and IRES. A table head-to-head comparison is highly recommended.

We added extra information about strategies, and also provided the Table comparing all of the described strategies (Lines 64-67, 71-74).

3. Some necessary schematic depicting 2A peptides and IRES should be provided.

In view of the fact that the IRES structure is already comprehensively described elsewhere and the 2A peptides are aminoacidic sequences and have not secondary structure we have not discussed these aspects in detail. However, we have inserted references to the published works in which the structure and aminoacidic sequences of IRES and 2A peptides are provided (Lines 105-107, 131-132).

4. In the section of “3. Self-cleaving 2A peptides”. Different 2A peptides sequence used in vector constructor should be summarized and compared.

We have broadened the part about different 2A peptides sequence (Line 162-181).

5. Some novel retroviral replicating vectors involved in gene delivery should be also overviewed here. As a hot topic, inclusion of some non-viral gene delivery vector, like nanoparticles-based strategy, will greatly broaden the spectrum of this review, which might cover more reader’ interests.

We have added information about strategies for delivery, including viruses and nanoparticles in Introduction (Line 43-46).

6. Under section 4 to 8, to attract reader’s attention, some figures with interesting results and important finding in those literatures can be listed, instead of only using plain text.

Some extra information from those articles has been added into the manuscript.

7. Multicistronic vectors for other diseases field should also be mentioned. Such as infectious diseases

A new part about the use of multicistronic vectors in the therapy of infectious diseases was added (Line 394-414).

8. Line 326: [81]. “The use of 2A peptides in cancer treatment so far is reduced to an increase in the production of”. There is a grammar error in this sentence. “so far is reduced due to” not “so far is reduced to”. Authors should verify on that.

The typo was corrected.

9. Abbreviation list should be included.

According to the Instructions for authors abbreviations should be defined in parentheses the first time they appear in the text, therefore sadly we cannot add a list.

Reviewer 2 Report

Manuscript ID: pharmaceutics-626016

The review article entitled “Production and application of multicistronic constructs for various human disease therapies” by Alisa Shaimardanova and co-workers is a survey about recent advances in the development and use of multicistronic vectors for the treatment of various human diseases, and specifically focused on internal ribosome entry site (IRES) and 2A nucleotide sequences.

Generally speaking, this review article has covered the topic in enough breadth and depth, and fits in perfectly with the scopes of Pharmaceutics journal, and equally well with the topics covered by special issue “Gene Delivery Vectors and Physical Methods: Present and Future Trends”.

Though this is a roughly interesting work, some issues and minor concerns deserved to be addressed before being re-considered for publication.

From the abstract onward, I suggest to replace 2A peptide with 2A system or 2A sequence when the authors specifically refer to a nucleotide sequence. Of note, 2A peptide is the translation product of a 2A sequence. The term peptide is used inappropriately even when comparing IRES nucleotide sequence with 2A. In this context, subheading 3 (Page 2, line 93) and the sentence from line 94 to 99 (Page 3) shall be revised. For the sake of clarity and by the same token, the statement “Application of a retroviral vector containing the amyloid precursor protein (APP), Tau protein (TAU) and human presenilin 1 (PS1) genes with six well-characterized mutations, separated by 2A peptide…” is formally incorrect.

Page 3, lines 105-109. As the text stands, the overall mechanism of action of 2A is somewhat unclear. Reference 21 (Liu et al., Sci Rep; 7: 2193. DOI: 10.1038/s41598-017-02460-2) and some other research papers and review articles provide clearer and greater mechanistic insight into 2A, and authors may benefit from reading these. Of note, about 2A-mediated Figure 1b displays ribosome skipping only. Figure 2b legend shall be revised accordingly.

Table 1 is well thought-out and displays very interesting information about IRES and 2A, but needs to be better embedded into the manuscript, that is better linked to the text. Besides, please revise the Table according to my suggestion: “after” and “before” the IRES shall be replaced with “downstream” and “upstream”.

Figure 1 shall display (a) and (b) according to the figure caption.

Some typos and a few repeated words shall be corrected, and some sentences shall be rephrased. For instance, please replace “several” with “many” (Page 1, line 41). Page 2, lines 89-90: what do you mean with “genes below the IRES”? Page 3,line 130: please replace “type of cell culture” with “cell type”. Page 8, line 342-343: the last sentence of the review just leaves a bit of a bad taste in the mouth, and defeats the purpose of the review itself. Authors shall emphasize any strength and weakness of multicistronic constructs and bring forward concrete ideas about what should be done next to foster their use in clinics.

Author Response

Thank you for your comments. We have corrected them point by point within the manuscript accordingly (your comments are in bold text and our responses are in ordinary type):

1. From the abstract onward, I suggest to replace 2A peptide with 2A system or 2A sequence when the authors specifically refer to a nucleotide sequence. Of note, 2A peptide is the translation product of a 2A sequence. The term peptide is used inappropriately even when comparing IRES nucleotide sequence with 2A. In this context, subheading 3 (Page 2, line 93) and the sentence from line 94 to 99 (Page 3) shall be revised. For the sake of clarity and by the same token, the statement “Application of a retroviral vector containing the amyloid precursor protein (APP), Tau protein (TAU) and human presenilin 1 (PS1) genes with six well-characterized mutations, separated by 2A peptide…” is formally incorrect.

We have revised all the text and corrected the name of 2A peptide where necessary.

2. Page 3, lines 105-109. As the text stands, the overall mechanism of action of 2A is somewhat unclear. Reference 21 (Liu et al., Sci Rep; 7: 2193. DOI: 10.1038/s41598-017-02460-2) and some other research papers and review articles provide clearer and greater mechanistic insight into 2A, and authors may benefit from reading these. Of note, about 2A-mediated Figure 1b displays ribosome skipping only. Figure 2b legend shall be revised accordingly.

We have extended the description of the mechanism of 2A peptide-based protein synthesis and also described the cases when the skipping is not sufficient. Some information was also added into the Figure legend (Lines 135-150).

3. Table 1 is well thought-out and displays very interesting information about IRES and 2A, but needs to be better embedded into the manuscript, that is better linked to the text. Besides, please revise the Table according to my suggestion: “after” and “before” the IRES shall be replaced with “downstream” and “upstream”.

We have improved Table 1.

4. Figure 1 shall display (a) and (b) according to the figure caption.

We have added (a) and (b).

5. Some typos and a few repeated words shall be corrected, and some sentences shall be rephrased. For instance, please replace “several” with “many” (Page 1, line 41). Page 2, lines 89-90: what do you mean with “genes below the IRES”? Page 3,line 130: please replace “type of cell culture” with “cell type”. Page 8, line 342-343: the last sentence of the review just leaves a bit of a bad taste in the mouth, and defeats the purpose of the review itself. Authors shall emphasize any strength and weakness of multicistronic constructs and bring forward concrete ideas about what should be done next to foster their use in clinics.

All the typos have been corrected and the Conclusion section was improved (Lines 426-453).

Reviewer 3 Report

The review article of Alisa A. Shaimardanova, Daria S. Chulpanova, Kristina V. Kitaeva, Ilmira I. Abdrakhmanova, Vladislav M. Chernov, Catrin S. Rutland, Albert A. Rizvanov and Valeriya V. Solovyeva as Co-authors: “Production and Application of Multicistronic Constructs for Various Human Disease Therapies” describes studies of multicistronic vectors based on IRES nucleotide sequence and self-cleaving 2A peptides, including its features and possible application for the treatment and modeling of various human diseases. Although the number of studies in this field is relatively small, the works described in the article have demonstrated promising data.

This was a well-described review.

Prior to publication, please address the following issues:

Please give a more detailed description of the aim of work, description of sub-paragraphs of review. In the present version, there is an only general sentence – “This review discusses the perspectives of the use of IRES and 2A peptides to create multicistronic vectors for gene and gene-cell therapy of various human disease.”
Please, in the conclusion part give short conclusions regarding the described topics.

Consequently, I do recommend accepting this manuscript for publication with minor revision.

Author Response

We appreciate the reviewer’s comments and thank them for their time. The following contains our point-by-point responses (your comments are in bold text and our responses are in ordinary type):

1. Please give a more detailed description of the aim of work, description of sub-paragraphs of review. In the present version, there is an only general sentence – “This review discusses the perspectives of the use of IRES and 2A peptides to create multicistronic vectors for gene and gene-cell therapy of various human disease.”

We have improved the Introduction section (Lines 85-88).

2. Please, in the conclusion part give short conclusions regarding the described topics.

The comprehensive description of all the sections was added into the Conclusion section (Lines 426-453).

Round 2

Reviewer 1 Report

Authors has already addressed most of my concerns and improved the quality of this manuscript, which is now accepted for publish.